# The 100 m Composite Ship?

**Michael John Lowde** [1,2,3], **Henry George Arthur Peters** [1], **Ruadan Geraghty** [1], **Jasper Graham-Jones** [1], **Richard Pemberton** [1] and **John Summerscales** [1,*]

1   Materials and Structures (MAST) Research Group, School of Engineering, Computing and Mathematics (SECaM), Reynolds Building, University of Plymouth, Plymouth PL4 8AA, UK; michael.lowde@students.plymouth.ac.uk (M.J.L.); henry.peters@students.plymouth.ac.uk (H.G.A.P.); ruadan.geraghty@plymouth.ac.uk (R.G.); jasper.graham-jones@plymouth.ac.uk (J.G.-J.); richard.pemberton@plymouth.ac.uk (R.P.)
2   Engineering Department, Yeovil College, Mudford Rd, Yeovil BA21 4DR, UK
3   Leonardo UK Limited, Lysander Road, Yeovil BA20 2YB, UK
*   Correspondence: j.summerscales@plymouth.ac.uk; Tel.: +44-1752-5-86150

**Abstract:** Fibre-reinforced polymer (FRP) matrix composites are widely used in large marine structures, and in wind turbines where blade lengths are now over 100 m. Composites are the material of choice for small vessels due to ease of manufacture, high hull girder stiffness, buckling resistance, corrosion resistance and underwater shock resistance. Ships over 100 m are still built using traditional steel and/or aluminium, but so far not FRP. Composite ship lengths have increased over the past 50 years, but fundamental technical challenges remain for the 100 m composite ship. Preliminary studies suggest a possible 30% saving in structural weight, a 7–21% reduction in full load displacement, and a cost saving of 15%. However, economic considerations, design codes, manufacturing limits, safety and end of life scenarios need to be addressed before a 100 m ship is built. Innovative materials and structures, notably carbon fibre composite skinned sandwich construction, or aramid fibres with vinylester modified epoxy resin, should result in increased mechanical performance and consequent improvements in economics and manufacturing processes. A linear extrapolation of length vs. launch dates predicts the first 100 m ship would be launched in 2042.

**Keywords:** carbon fibre; glass fibre; composite; framed single skin; monocoque; sandwich structure



## 1. Introduction

Fibre-reinforced polymer (FRP) matrix composites are widely used in the marine environment [1–4]. The advantages of FRP (relative to traditional shipbuilding materials such as wood, aluminium, and steel) include low densities, excellent modulus- and strength-to-weight ratios, corrosion resistance, and low maintenance requirements. Lighter hulls give improved fuel consumption and/or increased cargo capacity. The majority of composite vessels use woven glass fibre fabric to reinforce a thermosetting resin providing good strength, lightness, and manufacturability at a reasonable cost [5–7]. The construction may be a framed single skin, a monocoque thick skin or a sandwich structure. High-performance vessels may use carbon or aramid fibres, often hybridised with glass, to achieve better mechanical properties with reduced density, whilst retaining cost-effectiveness. Wahrhaftig et al. [8] reviewed the use of sandwich panels for marine structures.

The potential applications of composites include marine/offshore renewable energy (MRE/ORE) systems [9], oil and gas exploration and exploitation (OGEE) structures [10], dock infrastructure, submarines and submersibles, naval vessels, workboats, fast ferries, power boats, lifeboats, yachts, stern-gear (propellers and rudders), rigging: (wing-)masts and sails, leisure (canoes, kayaks, surfing, etc.) and aquaculture. Figure 1 shows some current state of the art composite structures drawn at the same scale.

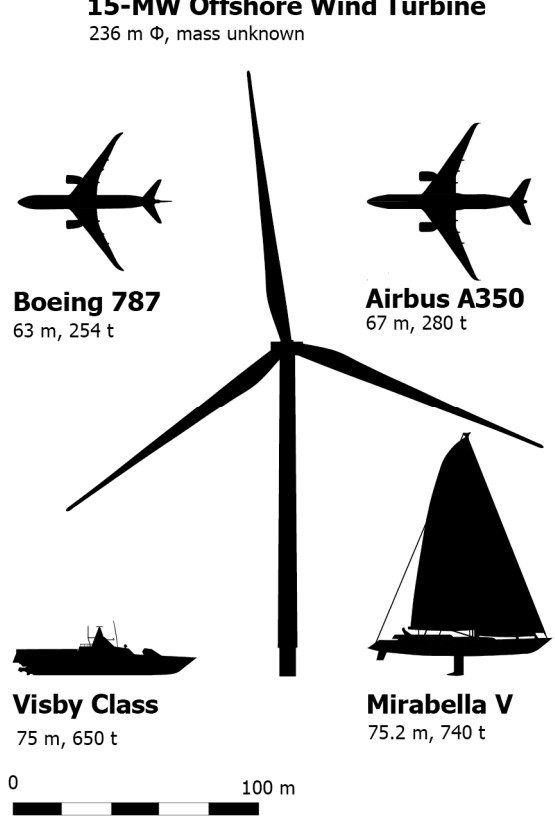

**Figure 1.** Large composite structures drawn to scale (a) Boeing 787 Dreamliner, (b) Airbus A350XWB, (c) Vestas V236 wind turbine with 115.5 m blades, (d) HSwMS Visby, (e) Mirabella V.

## 2. Large Composite Vessels

Composites are the material of choice for mine counter measures vessels (MCMV) due to their non-magnetic nature, underwater shock resistance, high hull girder stiffness, buckling resistance and corrosion resistance. The Royal Navy (RN, UK) constructed HMS *Wilton*, a 450 tonne (t) 46.3 m long monolithic Glass Reinforced Polyester (GRP) MCMV in 1973 with a **framed single skin** structure to the *Coniston* ("*Ton*") class wooden minehunter design [11]. The RN subsequently developed the 725 t 60 m long monolithic GRP *Hunt* class MCMV (13 in class) [12], then the 450 t 50 m long monolithic GRP *Sandown/Racecourse* class Single Role Mine Hunters (SRMH, 13 in class) [13]. A number of the ships are still in service, with vessel service life expected to exceed 50 years [14,15].

The 544 t 47 m Tripartite MCMV are contemporary with, and of similar construction to, the Hunt class vessels. The maximum (current) number of vessels were (are) Belgian Navy *Aster* class: 10(6) vessels, French Navy *Éridan* class: 14(10) vessels, Royal Netherlands Navy *Alkmaar* class: 15(6) vessels. Indonesia and Pakistan each acquired two vessels new. Decommissioned vessels were transferred to the navies in Bulgaria (3), Latvia (5) and Pakistan (1).

Russia claims the largest monolithic fibreglass hull for the 890 t 62 m *Alexandrit*-class minesweeper built at the Sredne-Nevsky shipyard in St. Petersburg. RTM-Worx flow simulation software (Polyworx BV, Nijverdal, The Netherlands) was used to model the infusion manufacture of the first vessel [16,17].

The Italian Navy, in collaboration with the Intermarine Shipyard, developed the 503 t 50 m *Lerici* class (4 vessels all delivered in 1985) MCMV, then the 672 t 51 m *Gaeta* class (8 vessels delivered 1992–1996) with a unique **monocoque** thick-skin GRP construction supported by appropriately positioned deck and bulkheads [18]. Table 1 lists the variants on the design supplied to other fleets [19]. Intermarine also built the 680 t 52.5 m Finnish Navy *Katanpää* class MCMV between 2007–2010.

**Table 1.** Intermarine Mine-Counter Measures Vessels (MCMV) supplied to foreign navies.

| Navy | Vessel Class | Vessels | Built/Delivered |
|---|---|---|---|
| Royal Malaysian Navy | Mahamiru | 4 | 1985 |
| Nigerian Navy | Ouhe | 2 | 1987–1999 |
| United States Navy | Osprey | 12 | 1991–1998 |
| Royal Australian Navy | Huon minehunter coastal (MHC) | 6 | 1994–2003 |
| Royal Thai Navy | Lat-Ya | 2 | 1999 |

The Scandinavian navies have used **sandwich** construction for the Royal Swedish Navy 360 t 48 m *Landsort* class minehunters (1981–1991), Royal Danish Navy 300 t 54 m *Flyvefisken*-class patrol vessels and the Norwegian Navy 375 t 54 m surface effect ship *Alta* class minesweepers and *Orsøy* class minehunters.

The Swedish Navy 650 t (fully equipped) 73 m long *Visby* class stealth corvettes used hybrid carbon/glass fabric reinforced rubber modified vinylester skinned PVC core sandwich construction. Missions for the multirole ship include naval combat, surveillance, mine laying and anti-submarine warfare. These roles are enabled by the lighter, strong, stealthy (electromagnetic interference shielding with low radar, infra-red, magnetic and acoustic signatures) and shock resistant composite design. Vessel speed is quoted as >35 knots. The hull was infusion manufactured as 60 m$^2$ panels [7,20–22].

The 1250 t 80 m Saab Kockums (SK) MCMV 80 (to potentially replace Sweden's *Koster* class and the Belgium and Dutch Navy *Tripartite* MCMV) will act as a mothership for remotely operated or autonomous mine-warfare systems equipped with interchangeable mission module containers to fit the role of a full-fledged offshore patrol vessel (OPV). "If a customer prefers a non-composite hull, Kockums can offer the vessel with a steel hull" [23]. The Royal Singapore Navy worked with SK to design the 1250 t 80 m *Independence* class Littoral Mission Vessel (LMV), but the design defaulted to a steel hull with a composite superstructure. SK have two larger corvettes at the planning stage: an 88 m Multi Mission Corvette and a 98 m FLEXpatrol [23].

A US Navy feasibility study for composite construction of their next generation corvettes concluded that for a 1200 t 85 m ship there could be a 30% saving in structural weight, a 7–21% reduction in full load displacement, and a cost saving of 15% relative to the equivalent steel vessel [21].

The Indonesian Navy produced a 219 t 63 m Fast Missile Patrol Vessel (FMPV) with a wave piercing trimaran design and helicopter platform constructed as a carbon fibre composite sandwich with vinylester modified epoxy resin matrix using a resin infusion process [24].

Carbon fibre composites have been used for the superstructures of larger vessels including (i) the Russian Navy *Admiral Gorshkov* class stealth frigate (circa 2012) [25], and (ii) the United States Navy *Zumwalt* class DDG 1000 destroyer upper-section deckhouse, helicopter hanger and ballistic screen [26,27].

The Ron Holland designed 740 t 75.2 m long *Mirabella V* single mast (sloop rigged) hyper-yacht was amongst the largest non-military composite vessels in the world. Built by Vosper Thornycroft and launched in 2004, with her 90 m mast stepped in late December 2004, materials of construction are carbon (deck and stiffeners), aramid (outside hull skin) and E-glass composites in a vinyl ester resin over PVC or polyolefin foam cores. Preimpregnated (pre-preg) materials were considered too labour intensive given the need to debulk every three to four plies. The vessel was hand laminated in a female tool with a Medium Density Fibreboard (MDF) surface supported by a timber framework [28,29]. In a 2013 retrofit at Pendennis Shipyard, Mirabella V became the 765 t 77.6 m long *M5* with a stern extension, new keel and rudder updates and her original steel rod rigging replaced with lighter carbon textile rigging [30].

Figure 2 shows some of the above vessels at the same scale.

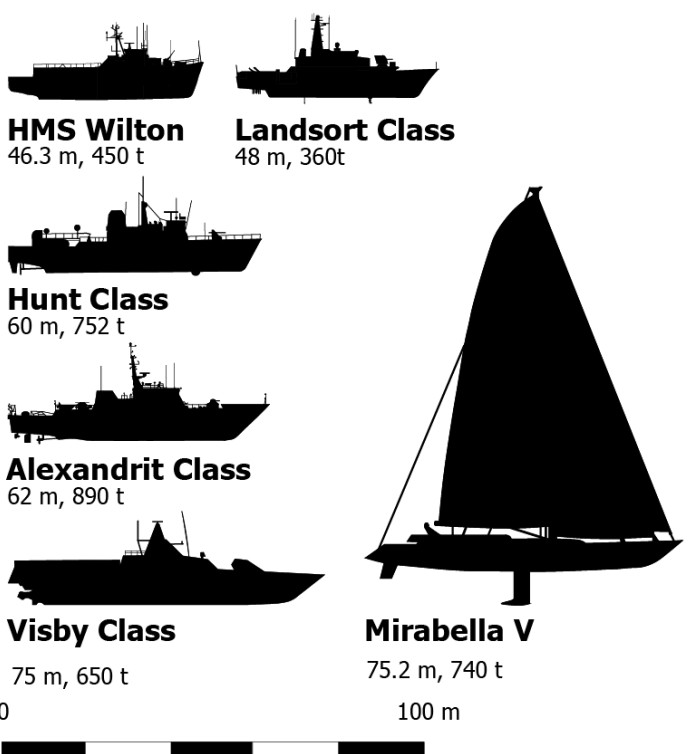

**HMS Wilton**
46.3 m, 450 t

**Landsort Class**
48 m, 360t

**Hunt Class**
60 m, 752 t

**Alexandrit Class**
62 m, 890 t

**Visby Class**
75 m, 650 t

**Mirabella V**
75.2 m, 740 t

0                                100 m

**Figure 2.** Representative composite ships and yachts drawn to scale.

Mouritz et al. [21] reviewed advances in FRP for naval applications driving research and development. The desirable characteristics of composites include improved range, and non-magnetic and stealth properties, making FRP the ideal choice for mid-sized MCMV, corvette and patrol vessels. They also suggested that the ~72 m length limitation arises from high costs and an absence of empirical performance data. Rubino et al. [7] reviewed recent applications of FRP in marine structures and agreed it is unlikely FRP will replace traditional materials for vessels between 80–160 m long. The principal constraints on size are high costs and the complex design process underpinned by unreliable performance prediction for FRP.

Mouritz et al. [21], Calabrese et al. [6] and Popham [31] all identified a maximum composite hull size below 100 m. The size limitations were linked to cost, technical, environmental, and manufacturing issues when compared to traditional shipbuilding materials, such as steel and aluminium, for which long-established data is available. The technical issues include unclear or poor guidance and legislation for design, safety, and end of life procedures, adding complexity and uncertainty to performance prediction. Definition of lifecycle and true environmental impact for composite hulls was complicated by high material costs and environmentally harmful, labour-intensive manufacturing processes in combination with a lack of empirical performance data.

## 3. Economic Considerations

A ship is normally a capital-intensive project. Any ship project should consider both the up-front acquisition cost, lifetime operations, repair and maintenance costs, and decommissioning and disposal costs. The shipping industry is risk averse with long experience in the use of steel and established supply chain relationships. Change will require disruptive technologies. Composites as raw materials are normally more expensive than traditional engineering materials on a weight-for-weight basis. They are more competitive on a volume-for-volume basis given the higher specific properties due to the lower density.

Bader [32] conducted a study to estimate costs for the manufacture of a simple component in different composite materials with different manufacturing routes. He concluded that carbon fibre reinforcement could provide a better economic solution than glass fibre.

Crawford et al. [33] compared cost models for resin infused or open moulded GRP boat hulls and found that the infusion hull was ~23% cheaper to manufacture but only with a strategic plan for technician training ahead of the migration to the new technology. Parts integration and full life cycle costing (LCC) can often compensate for increased initial costs [34–37].

Hagnell [38] proposed a technical cost model for composite manufacture, assembly and basic inspection for aeronautical and transport applications. For low annual volumes, the principal cost driver was material costs, while for large parts and slow processes, the secondary cost driver was tooling costs. Labour and investment in automation were also significant cost drivers.

A quantitative comparison of material costs is fraught with difficulties, not least that bulk purchases attract different discounts. The Performance Composites website [39] has steel at 0.5–1.0, fibreglass/polyester at 2.0–3.0 and carbon/epoxy at 9.0–20.0 (original data in US$/lb). Meijer [40] developed a cost price estimation model using data for the single-skin stiffened Alkmaar class MCMV. The production cost estimate for the hull of the hand lay-up ship was €123/kg (2014/15 prices), being a factor 25 increase between the raw materials and a fabricated component produced within a defence quality system.

Production of 100 m vessels is likely to be for a short production run: the costs of mould tools amortised over a few products could be critical to the economic success of the project. The production of high-quality vessels requires a temperature-controlled environment: there are few (if any) shipyards with facilities of a size appropriate to this task.

The reduced displacement when changing from a metal to a composite hull can reduce fuel costs or increase payload. When the superstructure of the cruise ship Norwegian Gem changed to composites, the payback time for fuel savings would have been 5.9 years, but the capacity to add extra cabins led to payback in 2.5 years [14].

## 4. Design Codes and Considerations

The normal material for the construction of large vessels is steel. On a volume-for-volume basis, the elastic modulus of composites manufactured with standard grade fibres cannot match the modulus of steel. For fibre composites, the elastic modulus can be calculated using the rule-of-mixtures (Equation (1)):

$$E_c = \eta_l \cdot \eta_o \cdot E_f \cdot V_f + E_m \cdot V_m \tag{1}$$

where $E_x$ = elastic modulus of the material, $V_x$ = volume fraction of the component, $\eta_l$ = fibre length distribution factor (FLDF), $\eta_o$ = fibre orientation distribution factor (FODF), and subscripts c, f and m indicate composite, fibre and matrix, respectively. The strength of unidirectional composites can be estimated using the Kelly-Tyson equation (Equation (2)):

$$\sigma_{c'} = \sigma_{f'} \cdot V_f + \sigma_{m*} \cdot V_m \tag{2}$$

where $\sigma_{x'}$ is the strength of the material and $\sigma_{m*}$ is the stress in the matrix at the failure strain of the matrix. For composites with fibres transverse to the load, a failure strain for fibre-matrix debonding might be set at 0.20–0.25%. However, composites materials are normally of lower density than metals, so specific properties (property/density) are used to compare the performance on a weight-for-weight basis. The matrix term in Equation (1) or Equation (2) has been assumed to be negligible. The properties in Table 2 assume densities ($\rho$ in kg/m$^3$) of 1200 (resin), 1800 (carbon fibre) or 2550 (glass fibre). Fibre moduli are 72 GPa (glass) or 235 GPa (carbon), and fibre strengths of ~3 GPa. FLDF is set at unity (1) for continuous fibres. A quasi-isotropic (QI) laminate has an equal distribution of fibres in four directions (0°, ±45°, 90°), a woven roving composite has fibres in two directions (0°, 90°), and a unidirectional composite has all fibres aligned with the stress (0°). Composite strengths are calculated assuming only fibres aligned with the reference axis carry stress and negligible contribution from the matrix.

**Table 2.** Comparison of material properties (CFRP = carbon fibre composites, GFRP = glass fibre reinforced composites, Al = 5083 aluminium, steel = 316 L grade, specific properties converted to display as integers).

|  | QI GFRP | WR GFRP | UD GFRP | QI CFRP | WR CFRP | UD CFRP | Al | Steel |
|---|---|---|---|---|---|---|---|---|
| Fibre volume fraction | 0.5 | 0.5 | 0.65 | 0.5 | 0.5 | 0.65 | N/A | N/A |
| Density ($\rho$ in kg/m$^3$) | 2078 | 2078 | 1875 | 1590 | 1590 | 1500 | 2650 | 8000 |
| FODF | 0.375 | 0.5 | 1 | 0.375 | 0.5 | 1 | N/A | N/A |
| Elastic modulus (GPa) | 13.5 | 18 | 47 | 44 | 59 | 153 | 72 | 193 |
| Specific modulus ($E/\rho$) | 5 | 7 | 25 | 24 | 37 | 102 | 27 | 24 |
| Strength (MPa) | 375 | 750 | 1950 | 375 | 750 | 1950 | 300 | 485 |
| Specific strength ($\sigma_{c'}/\rho$) | 180 | 361 | 1040 | 236 | 472 | 1300 | 113 | 61 |

Composites manufactured with standard grade fibres will not substitute for steel on a volume-for-volume basis. However, coal-tar pitch-based carbon fibres (e.g., Mitsubishi Rayon Dialead K13D2U grade) are available with moduli up to 935 GPa, albeit with low elongation at break (0.4%) and at high cost. The assumptions used for Table 2 would predict an elastic modulus greater than that of steel for a woven roving fabric composite using the pitch fibre.

New materials, for example carbon nanotubes, and graphene (and other two-dimensional monolayers) platelets are still mostly at laboratory scale and hence not yet available at sensible cost or in the quantities required for the construction of large structures.

The longitudinal bending strength of a ship is measured as the hull girder section modulus. For a composite structure to match the flexural performance of steel, material thickness would normally be increased. As the material is acting at a distance from the neutral axis, the required increase in mass would be lower than the ratio of the respective material moduli. In addition to this, the efficiency of composite structures has increased by utilising greater geometric complexity, integrating various structural elements and incorporating low-cost, high performance carbon fibre pultrusions, as has been seen in the wind turbine industry.

Unlike traditional engineering materials, fibre-reinforced composites are normally anisotropic. Further, the wide variety of ply stacking sequences available for composite structures gives the designer considerable flexibility in the achievable properties. However, that variety does mean that it is rarely possible to pull the elastic and strength properties from standard reference texts. Furthermore, current production routes produce materials with high levels of variability in both properties and quality. Manufacturing processes and conditions, and chemical composition can all affect the performance of the final material [41]. Mitigation measures might require high tooling investment, strong process control, and experimental validation of the systems.

The shipping industry prefers conservative empirical design from first principles as this is a cheaper, safer and risk averse approach. However, composites lack good design standards and regulations with conservative designs needing large factors of safety. Mouritz et al. [21] found that the factor of safety could be up to 10 for some composite panels in certain circumstances. Unlike the aerospace industry where weight saving is a primary driver, the shipping industry has to consider ship stability when lighter materials low in the vessel could adversely affect the centre of buoyancy [7,15,28,42]. The absence of comprehensive data on composites, and a need for experimental verification to validate novel designs, coupled with the low technology of the shipping industry, constrains the fundamental analysis required to produce a 100 m complex composite hull [28,43].

The prospects for large composite ships were the focus of the RAMSSES (EU Horizon 2020 grant agreement 723246 Realisation and Demonstration of Advanced Material Solutions for Sustainable and Efficient Ships, 2017–2021) and FIBRESHIP (EU Horizon 2020 grant agreement 723360 Engineering, production and life-cycle management for the complete construction of large-length FIBRE-based SHIPs, 2017–2020) projects. A joint presentation [44] noted that SOLAS Ch.II-2 Regulation 2 requires "The hull, superstructures, structural bulkheads, decks and deckhouses shall be constructed of steel or other equivalent

material" and SOLAS Ch.II-2 Regulation 17 has "Alternative design and arrangements" on basis of Equivalent Safety.

National classification society rules control the design and manufacture of marine vessels, although the respective publications are used across borders. The major societies include:

- American Bureau of Shipping (Houston, TX, USA),
- Bureau Veritas (Neuilly-sur-Seine, France),
- DNV (Det Norske Veritas, Bærum, Norway, and the former Germanischer Lloyd, Hamburg, Germany),
- Lloyd's Register (LR, London, UK), and
- RINA SpA (Registro Italiano Navale, Genoa, Italy).

Key documents in this context include guidance from Germanischer Lloyd [45], Lloyd's Register [46], International Maritime Organization (IMO) [47,48] and, specific to composites, DNV-GL [49] and IMO [50].

The FIBRESHIP3 Integrated Composite Ship project reviewed the engineering, production and life-cycle management for the complete construction of large-length fibre-based ships. The consortium WP4 [51] developed design guidelines for three vessel concepts (a light commercial container ship, a roll-on/roll-off passenger (ROPX) ferry and an oceanographic fishing research vessel) and assessed the technical implications of complete build in fibre-based materials. "The most important finding of this study is that the structural design of a vessel of length up to 100 m is technically feasible, considering the currently available resources in the vessel design and shipbuilding sector". Maccari [52] presented new guidance for the adoption of innovative techniques for building large vessels in composite materials, supplementing the provisions of the current regulatory framework.

Wahrhaftig [8] reviewed the application of FRP-skinned sandwich panels for marine structures. Flexure of panels for small- and mid-sized vessels can be restricted by internal shear webs that run transverse and longitudinally through the hull. The application of FRP skin sandwich structures to large marine vessels may require internal stiffening-supports increasing weight and reducing internal hull space.

## 5. Safety of Life at Sea Convention (SOLAS)

The SOLAS Convention [53], first adopted in 1914 after the RMS *Titanic* disaster, is regarded as the most important international treaty concerning the safety of merchant ships. The main objective of the convention is to specify minimum standards for the construction, equipment, and operation of ships, compatible with their safety. Sailors generally fear the three Fs: fire, fog (impeded vision) and frost (icebergs).

A fire on board a ship is one of the most dangerous scenarios it can encounter, especially if at some distance from coast guard/rescue services and a safe harbour. Fire (flame, smoke and toxicity/FST) can be a major design challenge for marine vessels. SOLAS requires that all structural materials are non-combustible and pass the ISO 1182 non-combustibility test. The first precaution is to avoid ignition whether from electrical issues, unstable chemicals and fuels, or enemy missiles. In the event of fire propagating there should be no/slow spread of flame and minimal emissions of toxic substances.

Mouritz et al. [21] assessed the post-fire mechanical properties of marine grade glass fibre reinforced polymers (GFRP) with polyester, vinyl ester or phenolic resin matrix systems. They found that phenolic resin had better resistance to heat degradation, but all composites had substantial loss in flexural and tensile properties due to degradation of the matrix material and delamination of the plies.

Two RN *Hunt* class MCMV have suffered major engine room fires (M30 *Ledbury* in 1983 and M31 *Cattistock* in 1997) but both are back in active service. While steel does not burn, it does conduct heat well and can lead to "flashover" fires in adjacent compartments. The engine room fire on HMS *Ledbury* burned for several hours but adjacent compartments did not require boundary cooling and suffered minimal damage [14].

The Swedish Lightweight construction applications at sea (LASS) project conducted fire tests using "real-world" fire scenarios to demonstrate that fire as a major obstacle to the wider use of composite structures in ships can be overcome from both technical and practical viewpoints. A composite system with 75 mm of 110 kg/m$^3$ mineral wool fire insulation developed for 60 min fire protection was >15% lighter than "A" class steel bulkheads [54].

In 1988, 167 lives were lost when the Piper Alpha OGEE offshore platform was destroyed by an onboard explosion and hydrocarbon fires. Offshore composite structures are now required to have fire protection for up to 2.5 h against Hydrocarbon (H-Class) and Jet (J-Class) fires. Composites with multiple ablative layers achieve heat flux up to 350 kW/m$^2$ [55] by pyrolysis of surface material to leave a porous char layer, which provides additional insulation for the remaining solid material [56].

The FIBRESHIP project identified that FRP materials in vessels above 50 m length are only allowed for secondary structural elements of the vessel [57]. The project sought to create and promote new regulatory frameworks to enable composite laminate construction of structures in longer vessels primarily for weight reduction relative to conventional steel ships. Lower vessel weight should increase cargo capacity, decrease vessel fuel consumption and emissions, and reduce corrosion and consequent maintenance costs. Laboratory scale experimental fire performance studies characterised the candidate materials to understand their fire behaviour, including thermal degradation behaviour, smoke production, and toxicity. Coupled computational fluid dynamics (CFD) and finite element analysis (FEA) simulations of fire scenarios used standard time-temperature curves, and realistic fire descriptions, to assess the temperature dependence and thermal degradation of the composite through to global structure behaviour and collapse during a fire event.

VTT (Technical Research Centre of Finland) conducted a small-scale fire testing campaign on seven composite material systems (Table 3) to evaluate the time to ignition (TIG), maximum (HRRmax) and total (THR) heat release rates (HRR) and total smoke production (TSP) using a cone calorimeter. The phenolic resin system had the best fire performance for time to ignition, heat release and smoke production. The three epoxy resin systems showed similar TIG behaviour, but high heat release and smoke production. The acrylic polymer had the shortest time to ignition, intermediate heat release rates and low smoke production. The laminates with intumescent surface had good-reaction-to-fire performance with a long ignition time and low heat release rate and smoke production. The vinylester system had acceptable mechanical properties and superior fire performance and was selected as the potential resin for laminates used in the new vessel designs. FIBRESHIP also performed real-case fire simulations in different fire scenarios to understand the role of the composite materials and the fire location in fire propagation.

**Table 3.** Fire performance of the FIBRESHIP composite laminates ranked by decreasing time to ignition (TIG) [57,58].

| Polymer | System | Supplier | TIG (s) | HRRmax (kW/m$^2$) | THR (MJ/m$^2$) | TSP (m$^2$) |
|---|---|---|---|---|---|---|
| phenolic resin | Cellobond$^{TM}$ J2027X | Hexion | 101 | 71 | 9.9 | 0.4 |
| vinyl ester | LEO system with(out) topcoat | Saertex | 75 (50) | 69 (336) | 42.3 (33.5) | 8.8 (15.1) |
| bio-based epoxy | Super Sap$^®$ CLR | Entropy | 61 | 520 | 42.0 | 12.0 |
| epoxy resin | Prime$^{TM}$ 27 | Gurit | 60 | 496 | 39.4 | 10.7 |
| epoxy resin | SR1125 with(out) SGi 128 intumescent gelcoat | Sicomin | 52 (53) | 261 (546) | 40.7 (42.5) | 9.3 (13.5) |
| urethane acrylate | Crestapol$^®$ 1210 | Scott Bader | 44 | 314 | 35.4 | 9.3 |
| methacrylic | Elium$^®$ thermoplastic | Arkema | 23 | 255 | 40.7 | 1.8 |

Pacheco-Blazquez et al. [59] modified the Serial/Parallel Rule of Mixtures (SPROM) composite constitutive model to include the effect of thermal expansion in the context of the FIBRESHIP project. A 1D through-thickness thermal model with pyrolysis based on the classic Henderson experiment [60] was used to obtain an initial temperature profile. A four-node quadrilateral flat shell element (QLLL: quadrilateral and linear deflection, rotation

and shear fields) [61] was then used in the thermo-mechanical model. The numerical simulation was validated against an original vertical furnace test of an FRP ship bulkhead following International Code for Application of Fire Test Procedures (2010 FTP Code Part 11) standards [62].

## 6. Manufacturing Considerations

Normally manufacture of composite marine vessels takes place in a mould tool [63]. For a very large vessel, the tooling costs amortised over small production numbers may make the project financially unviable. Composite vessels may be manufactured by open mould gel-coating then spray-, or hand-, lamination processes, but health and safety considerations around exposure to volatile organic compounds (VOC, especially styrene) require expensive investment in workspace ventilation [64].

The processes of choice for larger vessels have become resin infusion under flexible tooling (RIFT, also known as SCRIMP™ or VARTM) [6,65–73] for leisure craft, or pre-impregnated materials within a vacuum bag for racing vessels. The bagging processes provide some consolidation of the laminate giving higher fibre volume fractions and reduced resin-rich volumes [74] and associated voids [41], and hence better performance, than the open mould methods. Even for medium-sized FRP marine vessels, correct infusion and consolidation across the entirety of the laminate requires significant time, labour and cost, and failure can lead to both financial and environmental ramifications through the wasted time, material, energy and resources. For larger vessels, limits on successful manufacture include workforce skills, manufacturing conditions, an enclosed temperature-controlled facility, and enhanced process controls.

As a vessel becomes longer, the keel (the lowest main beam) to gunwale (upper edge of the vessel's side) distance generally increases, so infusion to height becomes a constraint. Polyworx BV (Nijverdal, The Netherlands) worked with the SNSZ shipyard (St. Petersburg, Russia) towards infusion of a 62 m long by 10 m tall minesweeper hull. In theory, vacuum can only raise a column of resin to 9.5 m, but "for every meter of height, you lose 100 mbar in vacuum". The hull was infused in two steps, first from the keel to the chine [where the bottom and side meet], then from the chine to the gunwale [16]. The infusion required 21 t of resin, 45 t of fabric and 1.5 km of spiral feed tube with an 85 m × 35 m vacuum bag [75].

Critchfield et al. [76] considered low-cost design and fabrication of single-skin stiffened, monocoque and sandwich composite ship structures subject to air blast and shock. Components made using VARTM (a RIFT variant) generated mechanical properties comparable to wet lay-up or prepreg with autoclave processes.

Calabrese et al. [6] reviewed the manufacture of marine composite sandwich structures. Joulia and Grove [77] have presented initial performance data for "cut-and-fold" joints in honeycomb sandwich panels which may provide a route to more secure skin-to-core bonding for mould tools or components that can tolerate the developed geometry.

Qin et al. [78] have reviewed monomer selection for in situ polymerisation (ISP) during monomer infusion under flexible tooling (MIFT) manufacture of natural-fibre reinforced thermoplastic-matrix marine composites. Arhant and Davies [79] have reviewed the current status of materials, manufacturing methods, and durability of thermoplastic matrix composites for marine structures. The thermoplastic matrix would permit end-of-life treatments at a higher level in the waste hierarchy.

## 7. Life Cycle Assessment (LCA) and End-of-Life (EoL) Scenarios

In respect of durability, Harris [80] has considered the fatigue response of FRP while Pritchard [81], Martin [82], and Davies and Rajapakse [83] address the durability and ageing of reinforced plastic composites.

Well-manufactured composite hulls do not corrode and hence require less protective paint. This leads to less time out of the water for maintenance, longer periods at sea, increased vessel life and more revenue in a commercial setting. The causes of osmosis and

blistering are now well understood [84] and this degradation mechanism should not be a constraint to future vessel build.

Any product made from composites should undergo a life cycle assessment (LCA) to minimise its impact on the environment. Burman et al. [85] conducted a life cycle assessment for a high-speed patrol vessel. Their findings were that the most significant environmental impacts were abiotic depletion, global warming, and acidification, primarily arising from the burning of fossil fuel during the boat service life. Reduced displacement for a carbon fibre composite vessel gave a 20% reduction in fuel consumption relative to an aluminium vessel. However, vessel displacement does depend on cargo, crew and passengers (or for naval operations on weapons systems and ordnance).

Marine vessels are designed and built to be durable in a harsh environment. However, at end-of-life that creates a waste stream that is difficult to handle. Disposal should consider the waste hierarchy: reduce > reuse > repair > recycle > recover > incineration with energy recovery > incineration without recovery > dispose (landfill or scuttle). In the shadow of the End-of-Life (EoL) Vehicle (ELV) (Directive 2000/53/EC) [86] and Waste from Electrical and Electronic Equipment (WEEE) (Directive 2012/19/EU) [87], manufacturers of marine vessels increasingly need to prepare for Extended Producer Responsibility obligations [88] and, in due course, future regulations.

Whereas metal structures can be melted and recycled with relative ease [89,90], the high durability of composites restricts the end-of-life options. The long lifespan of composite vessels can result in multiple owners throughout life. The monetary costs of landfill and limits on incineration increase the risk of abandoned small and medium composite vessels ending up in geographically diffuse locations.

The use of thermosetting resin matrix systems severely limits the disposal options for both production waste and EoL composites, as reviewed by many authors [91–94]. Summerscales et al. [95] reviewed the disposal options for composite marine vessels. Parallel reviews [96–99] consider the EoL issues for composite wind turbine blades.

**Reuse:** Vessels from the RN MCMV fleet have been transferred to the Greek Navy (2) and the Lithuanian Navy (3), repurposed as a training ship (HMS *Brecon* was decommissioned and is now at HMS *Raleigh*) or as a clubhouse (HMS *Wilton* is now the home of the Essex Yacht Club).

After 24 years in service against a planned operational lifespan of 25 years, the RNLI established that the *Severn* lifeboat composite hulls can continue in operational service for another 25 years [100]. The life extension programme includes new shock-mitigating seats for the crew, new survivor space seating for casualties, and a new daughter craft for rapid recovery of casualties in shallow waters or close to rocks. Redundant RNLI lifeboats are sold worldwide as a revenue stream for the charity.

The most recent *Shannon* class RNLI offshore lifeboat has a 60 year design life before disposal, and anticipates new propulsion through easier engine changes. The *Shannon* will gradually replace the *Mersey* and *Tyne* class lifeboats, which are now close to the end of their operational lives. On completion of the roll out, the entire RNLI all-weather lifeboat fleet will be capable of 25 knots, to make the lifesaving service more efficient and effective than ever before [101].

When a ship becomes economically unviable and cannot be re-used, the composite materials can be re-purposed into lower duty structures such as jetties, bridge components and street furniture.

**Recycle:** The most common methods for recycling FRP are by mechanical, thermal, or chemical means. These processes generate recyclate as filler or particulate reinforcement (for polymer, asphalt or cement composites). The high recycling costs and low recyclate value, combined with a lack of established markets, are a poor incentive for composite users to pursue recycling [91,102]. Significant research and development is still required to develop practical and economic composite recycling.

**Recover:** Tertiary recycling methods use incineration, pyrolysis or solvothermal processes to break down and remove the resin matrix and expose the fibres. Pyrolysis tech-

niques can produce chemicals derived from the matrix as potential feedstock for future polymers. Thermal removal of the matrix severely degrades glass fibre strengths, although Thomason et al. [103] have demonstrated that short hot alkali treatment of recycled glass fibres can restore their ability to act as an effective reinforcement in second life composite materials.

**Dispose:** This method unfortunately is the most frequently used via abandonment, landfill or scuttling. Small GRP vessels are of comparable size to a car and can be stripped of valuable materials before crushing but the composite materials may still end up in landfill. Contamination with fuel, lubricants, paint and/or anti-fouling may constrain disposal options. Önal and Neşer [102] conducted a life cycle assessment to compare three EoL options (extruding, incineration or landfill) for GRP recreational boat hulls manufactured by hand lamination or infusion. The environmental impact of infusion was higher due to the energy consumption, but with lower risk in terms of occupational health.

### 8. Launch Date for a 100 m Ship?

So when might the first 100 m composite ship be launched? Figure 3 plots a limited selection of first launch dates for large composite vessels. A speculative linear extrapolation (Excel Trendline) indicates that the 100 m ship may meet the water late in the year 2041. Appendix A presents the data for vessels of wood or steel construction.

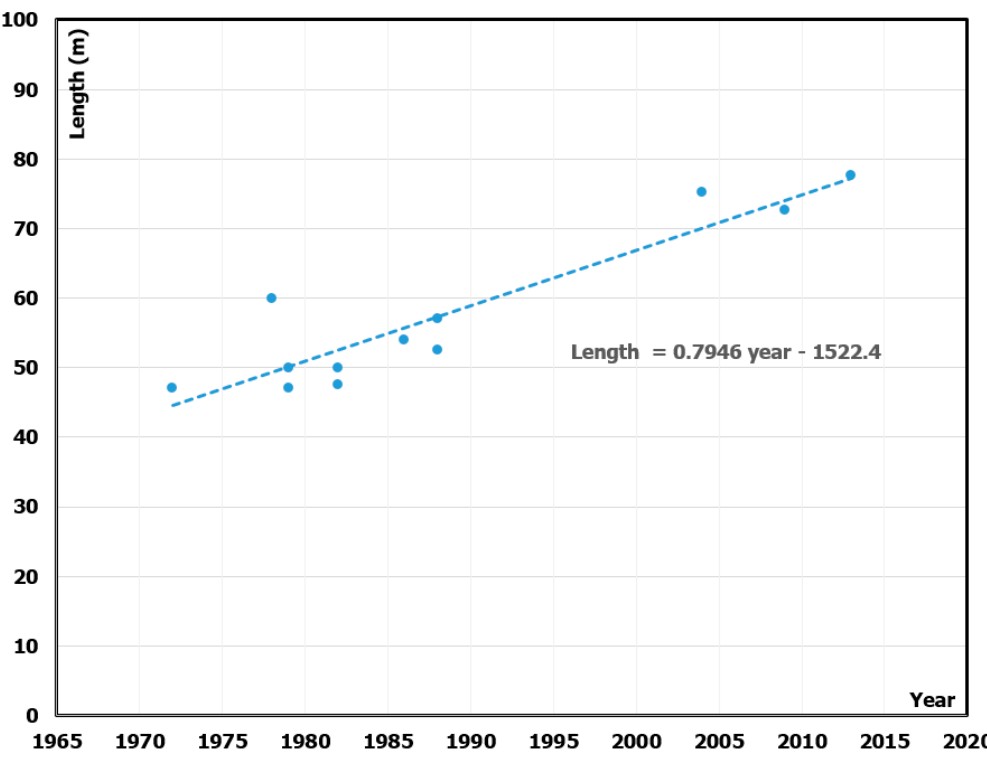

**Figure 3.** Projection of ship length against first in class launch date for key vessels.

### 9. Conclusions

The prospect of the 100 m overall length composite ship is constrained by a number of factors, with no immediate expectation of realisation in the short term. Key issues to overcome include:

(i)    the shipping industry prefers conservative empirical design from first principles as this often incurs lower up-front costs, and is a safer and risk averse approach,

(ii)    the absence of a comprehensive database of composite properties for the wide variety of candidate reinforcements and laminate stacking sequences,

(iii)   new materials (nanotubes, two-dimensional monolayers, etc.) to become available at sensible cost and in the quantities required.

(iv)   manufacturing routes to produce defect-free large structures,

(v)   life cycle costing and life cycle assessment to demonstrate the relative economic and environmental merits of the potential systems, and

(vi)   effective end-of-life technologies (especially if thermosetting matrices remain the matrix systems of choice).

Commercial composite wind turbine blades already exceed 100 m overall length. The 100 m ship will be a major turning point for composites in the marine sector and for similar complex structures. The FIBRESHIP project has demonstrated that the 100 m composite ship is technically feasible. The economic case for such a vessel will require life cycle thinking, and relaxation of the conservative attitudes of naval architects.

**Author Contributions:** Conceptualization, J.S.; Investigation, M.J.L. and H.G.A.P.; Writing—Original Draft Preparation, M.J.L. and H.G.A.P.; Writing—Review & Editing, M.J.L., H.G.A.P., R.G., J.G.-J., R.P. and J.S. All authors have read and agreed to the published version of the manuscript.

**Funding:** This research received no external funding.

**Institutional Review Board Statement:** Not applicable.

**Informed Consent Statement:** Not applicable.

**Acknowledgments:** The authors thank Howard Bradfield, Alvaro Oliviera, Ned Popham, and Gonzalo Diez Sanmartin for comments in the LinkedIn discussion of this topic.

**Conflicts of Interest:** The authors declare no conflict of interest.

## Appendix A

**Table A1.** A selection of historic, or large recent, ships ranked by length.

| Vessel | Purpose | Dates | Operator | Registered Tonnage | Length (m) | Beam (m) | Height/Draft (m) |
|---|---|---|---|---|---|---|---|
| Viking longship | longship | 850 | - | 20 | 30 | 8 | 30/45 |
| Golden Hind (was Pelican) | galleon | 1577 | (Queen Elizabeth I) | 300 | 31 | 6 | -/2.7 |
| Mayflower | transport | c.1609 | Christopher Jones | 180 | 32 | 8 | /4.0 |
| Olympias (trireme) | warship | 1987 | Hellenic Navy | 47 | 37 | 5.5 | -/1.25 |
| Mary Rose | warship | 1511 | Mary Rose Museum | 500/600 | 45 | 12 | -/4.6 |
| Vasa/Wasa | warship | 1627 | Wasa Museum | 1210 | 69 | 12 | 53/4.8 |
| HMS Victory | Gun ship | 1765 | Royal Navy | 3556 | 69 | 16 | -/8.8 |
| Bretagne | Gun ship | 1855 | (scrapped 1880) | 5289 | 120 | 18 | -8.56 |
| Solano | paddle steamer | 1878 | (scuttled 1931) | - | 130 | 35 | - |
| Wyoming | schooner | 1909 | (foundered 1924) | 3731 | 140 | 15 | -/9.3 |
| SS Great Britain | steamer | 1843 | Great Western Steamship | 3674 | 98 | 15 | -4.88 |
| Bismarck | battleship | 1939 | German Navy | 41,700 | 242 | 36 | -/9.3 |
| HMS Ark Royal | carrier | 1951 | Royal Navy (until 1979) | 36,800 | 245 | 34 | -/10 |
| RMS Titanic | liner | 1911 | White Star Line | 46,328 | 269 | 28 | 53/10.5 |
| USS Zumwalt DDG 1000 | destroyer | 2016 | United States Navy | 14,798 | 183 | 25 | -/8.4 |
| HMS Queen Elizabeth | carrier | 2017 | Royal Navy | 65,000 | 280 | 39 | -/11 |
| USS Nimitz class | carrier | 1975 | United States Navy | 106,300 | 333 | 41/77 | -/11.3 |
| Costa Smeralda | cruise | 2019 | Costa Cruises | 185,010 | 337 | 42 | -/8.8 |
| Independence of t'Seas | cruise | 2008 | RoyalCaribbean Freedom | 155,889 | 339 | 39/56 | -/8.53 |
| Iona | cruise | 2020 | P&O | 184,098 | 344 | | |
| RMS Queen Mary 2 | cruise | 2004 | Carnival/Cunard | 149,215 | 345 | 41/45 | -/10.3 |
| Oasis of the Seas | cruise | 2009 | Royal Caribbean Oasis 1 | 226,838 | 360 | 47/61 | 72/9.3 |
| Wonder of the Seas | cruise | 2022 | Royal Caribbean Oasis 5 | 236,857 | 362 | 47/64 | -/9.14 |
| HMM Algeciras | container | 2020 | Meritz Taurus | 228,283 | 400 | 61 | -/15.3 |
| Ever Ace | container | 2021 | Evergreen | 235,579 | 400 | 62 | -/17 |
| Mont (was Knock Nevis, Jahre Viking, Happy Giant, Seawise Giant) | tanker | 1979–2009 | Amber Development | 260,941 | 458 | 69 | -/24.6 |

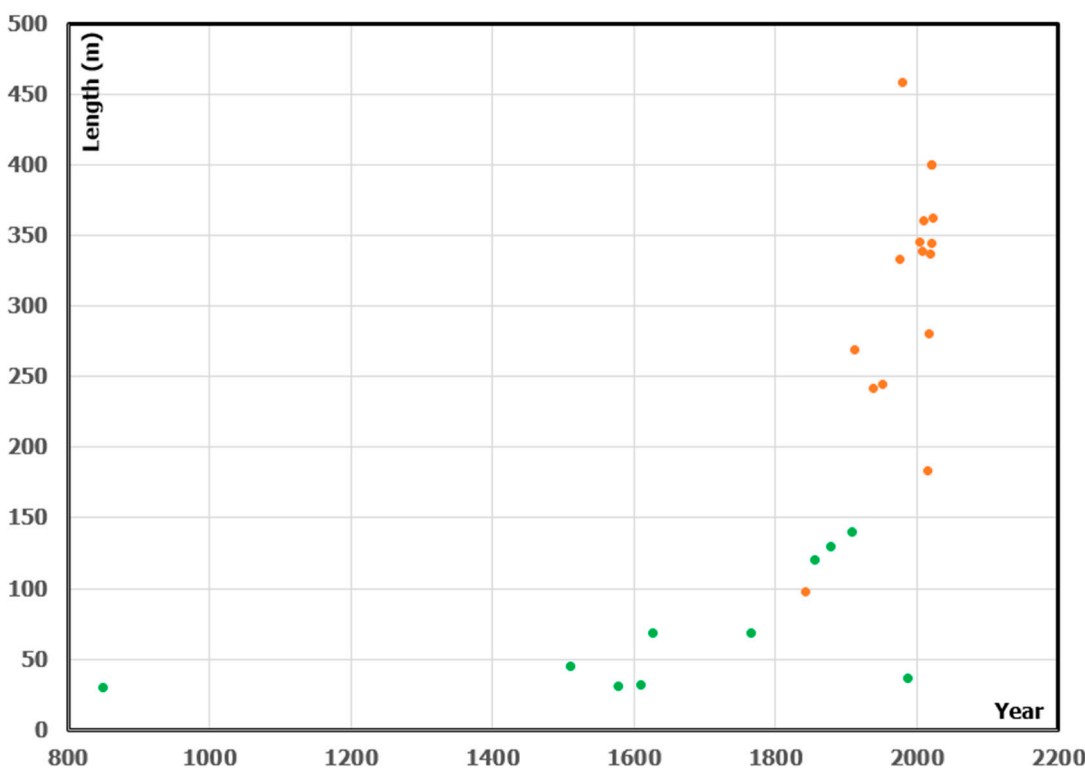

**Figure A1.** Ship lengths for wooden (green) and steel (orange) vessels by year.

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
