# Peer review of "The 100 m Composite Ship?"

_jmse, doi:10.3390/jmse10030408_

Round 1

Reviewer 1 Report

This is a paper review on the use of composite materials mainly for navy vessels in the last decades and future applications. The review by literature reference covers the main issues concerning composite materials application for shipbuilding: large composite vessels (still under 100 m length), economic, design codes, manufacturing, the safety of life at sea convention, life cycle assessment and end-of-life scenarios, prediction for 100 m length composite materials vessels launching in operation.

Reviewer 2 Report

This paper is precisely one of the article required for a special issue of Frontiers in ship and offshore structures as it provides an overview of advances in the application of composite materials and technologies in process of transition from small to large composite shipbuilding. This information is useful and necessary for the designer in the preliminary design phase, when much more than in steel (large) shipbuilding, should take into account the choice of available materials and associated production technology related to strength and safety requirements over construction. Through paragraphs It addresses all major project activities such as main dimension considerations, economic considerations, design code considerations, manufacturing considerations and life cycle assessment.

The title is well chosen and intriguing to read. Overall it is well structured. Number and merits of references are adequate. I can recommend publishing, only some small comments should be considered but not required:

  1. However, although it is a review paper, and yet it is about structures that is intriguing in its main dimensions and production process (which is largely indicated, somewhere in detail) it will be interesting to have some detail of the conceptual or classification documentation of the referenced structures, such as part of the general arrangement plan, main frame drawing or some structural detail, of the ships shown in the reference. Of course, bearing in mind how hard is to obtain this information and have the right to publish since these are projects that are protected by various forms of copyright because they are about advanced materials, production processes and consequently smart structural solutions. Still, this is exactly what the freshman as well as more experienced reader / shipbuilder / naval architect expects.
  2. As in 1. paper addresses relevant and actual EU projects such as FIBERSHIP which accomplished interesting results already. Maybe few more words or better one figure, as thousand words, can be shown.

So, within Chapter 2 or 4, if possible, insert few figures on structural detail from the actual construction or project.

Reviewer 3 Report

Dear Authors,

General comments:

Overall the article is a good read and contains valuable content. The article discusses the use of FRP as a material for the construction of large ships. Chapters examine important aspects of economics, design, manufacturing, safety and security and a life cycle assessment. As a whole, the article is a comprehensive review of the literature. Self-research is not included. The paper does not contain any novelties. I recommend publication as a review paper after major revision according to the comments listed as 'major issues'.   

Major issues:

  1. The abstract should be corrected to clearly define the aim and scope of the paper. At present the reviewer. As it stands, the reviewer, after reading the abstract alone, is unable to determine what is a general description and what is the stated content in the article. The repetition should be avoided. E.g. it says 'and a cost saving of 15%' and in the next sentences the authors again ask about the possible impact of PRF on production costs. Please shorten the abstract. Recommend focusing of what was done by the authors and what are the conclusions.
  2. The article lacks a chapter on the mechanical properties of composite materials in comparison with steel and aluminum alloys. What is a change of mechanical properties when temperature rises? The issue of lack of longitudinal hull stiffness made of composites should be discussed. Material itself has much lowest young modulus in comparison to steel. Similar issue is vital to aluminum alloys hulls.
  3. The conclusion section should be supplemented by the important considerations discussed in the preceding chapters. I suggest adding possible directions of development of materials and structures towards applications in large ships.
  4. Literature review has too many self-citations (>25%?!). I believe not all references are needed. On the other hand, there are some important references missing for new structural solutions between standard steel and composite (i.e. FRP) hull structure. I am referring to the steel sandwich panels, which are made of very thin steel sheets (2-4 mm). Please also see "steel sandwich panels with CFRP overlay" to reduce the fatigue cracking problem of thin plates. It may also be interesting to see the effect of lightweight concrete core material on the strength properties of hybrid sandwich panels to solve the fire resistance problem in a cost-effective manner. Steel sandwich panels may be a "bridge" to the use of a composite hull with fire-resistant escape routes made of hybrid structures.

Minor issues:

  1. Line 2: In title ‘100m’ should be ‘100 m’.
  2. Line 33: ‘excellent modulus’. It is not true in comparison with steel.
  3. Line 147: the decommissioning cost is also important
  4. Manufacturing considerations may include production of large composite ship using various materials and technologies integrated into a singe hull.
  5. The manufacturing section may include a sentence about the need of reuse of materials.
  6. Line 265: the reference [62] should not be in a section title. Put it into line 266.
  7. The section ‘4. Design codes and considerations’ should be supplemented by the major issue regarding design of non-steel hulls (also reflects to aluminum alloys) – fire resistance. I suggest to merge sections 4 and 6.
  8. I encourage you to add steel ship data in Figure 3. It would also be useful to supplement the chapter text with a comment about the increasing size of steel ships.

Kind regs,

Reviewer

Round 2

Reviewer 3 Report

As a whole, the article is a comprehensive review of the literature to discuss possible use of the FRP as a material for the construction of a 100 meter long ship. Self-research is not included. The abstract should be corrected to clearly define the aim and scope of the paper and indicate that it is a review paper. Without correction, it can also be published with little loss to the quality of the overall publication. The paper does not contain any novelties. I recommend publication as a review paper. 

Reviewer 4 Report

The manuscript has been modified in proper manner. It can be published in present form. 

A minor typo-errors are found, for instance, in new line 274, "A" is missing in ROPAX. They can be solved in proofreading stage.